# Pinpointing the tumor-specific T cells via TCR clusters

**Mikhail M Goncharov[1,2], Ekaterina A Bryushkova[2,3,4], Nikita I Sharaev[1], Valeria D Skatova[2,4], Anastasiya M Baryshnikova[2,4], George V Sharonov[4,5], Vadim Karnaukhov[1,2], Maria T Vakhitova[2,3], Igor V Samoylenko[6], Lev V Demidov[6], Sergey Lukyanov[4], Dmitriy M Chudakov[1,2,4]*, Ekaterina O Serebrovskaya[2,4,5]**

[1]Center of Life Sciences, Skolkovo Institute of Science and Technology, Moscow, Russian Federation; [2]Genomics of Adaptive Immunity Department, Shemyakin and Ovchinnikov Institute of Bioorganic Chemistry, Moscow, Russian Federation; [3]Department of Molecular Biology, Moscow State University, Moscow, Russian Federation; [4]Pirogov Russian National Research Medical University, Moscow, Russian Federation; [5]Laboratory of Genomics of Antitumor Adaptive Immunity, Privolzhsky Research Medical University, Nizhny Novgorod, Russian Federation; [6]Oncodermatology Department, NN Blokhin Russian Cancer Research Center, Moscow, Russian Federation

**\*For correspondence:**
chudakovdm@gmail.com

**Abstract** Adoptive cell transfer (ACT) is a promising approach to cancer immunotherapy, but its efficiency fundamentally depends on the extent of tumor-specific T cell enrichment within the graft. This can be estimated via activation with identifiable neoantigens, tumor-associated antigens (TAAs), or living or lysed tumor cells, but these approaches remain laborious, time-consuming, and functionally limited, hampering clinical development of ACT. Here, we demonstrate that homology cluster analysis of T cell receptor (TCR) repertoires efficiently identifies tumor-reactive TCRs allowing to: (1) detect their presence within the pool of tumor-infiltrating lymphocytes (TILs); (2) optimize TIL culturing conditions, with IL-2$_{low}$/IL-21/anti-PD-1 combination showing increased efficiency; (3) investigate surface marker-based enrichment for tumor-targeting T cells in freshly isolated TILs (enrichment confirmed for CD4$^+$ and CD8$^+$ PD-1$^+$/CD39$^+$ subsets), or re-stimulated TILs (informs on enrichment in 4-1BB-sorted cells). We believe that this approach to the rapid assessment of tumor-specific TCR enrichment should accelerate T cell therapy development.

## Editor's evaluation

This manuscript presents a computational approach to identify T-cells that can mount an immune response against tumors. The authors examine the presence of clusters of T cells with similar sequence as a surrogate for tumor antigen-specific responses. The identification of tumor-specific responses within the background of bystander T cell infiltration is an area of great current interest. This study provides solid support that T cell sequence clustering can be used to identify tumor-specific responses in vivo and in vitro.

## Introduction

Tumors develop diverse mechanisms of immune evasion, including the generation of hypoxic conditions (*Petrova et al., 2018*), inflammation (*Grivennikov et al., 2010*), establishment of an immunosuppressive microenvironment (*Ziani et al., 2018*; *Binnewies et al., 2018*; *Vijayan et al., 2017*; *Moesta et al., 2020*), downregulation of antigen presentation (*Kvistborg and Yewdell, 2018*; *Seliger et al.,*

2000; *Spranger et al., 2015*), promotion of regulatory T cell (T$_{reg}$) infiltration (*Curiel et al., 2004*) and outgrowth (*Adeegbe and Nishikawa, 2013*), and induction of T cell dysfunction (*Thommen and Schumacher, 2018*). The infusion of large numbers of expanded autologous tumor-reactive T cells—typically after the implementation of lymphodepleting regimens—represents a powerful therapeutic option that may override these immunosuppressive mechanisms. Clinical protocols for such adoptive cell transfer (ACT) therapeutic strategies (*Klebanoff et al., 2005*), inspired by the pioneering work of Steven Rosenberg's group (*Spiess et al., 1987*; *Rosenberg et al., 1994*; *Rosenberg et al., 1988*), are now being actively developed and used to treat patients (*Sarnaik et al., 2021*; *Dudley et al., 2003*; *Chapuis et al., 2016*; *Kelderman et al., 2016*; *Rosenberg and Restifo, 2015*).

There is accordingly great demand for methods for the enrichment of autologous tumor antigen-specific T cells for use in ACT protocols. Current techniques rely on the identification of patient-specific peptide neoantigens, which are then used for the functional characterization and selection of cultured tumor-infiltrating lymphocytes (TILs) (*Yossef et al., 2018*; *Leko et al., 2019*). Alternatively, since the identification of unique neoantigens is costly, time-consuming, and functionally limited in terms of the spectra of identifiable antigens, cultured autologous tumor tissue can be used as a source of antigen-specific stimulus (*Seliktar-Ofir et al., 2017*). Certain cell-surface markers of ongoing and chronic activation such as PD-1, CD39, CD69, CD103, or CD137 may also help to delineate T cell subpopulations (typically CD8$^+$) that are enriched for clonally expanded tumor-reactive T cells (*Gros et al., 2014*; *Duhen et al., 2018*; *Murray et al., 2016*), therefore culturing selected TIL subsets, such as PD-1$^+$ T cells (*Inozume et al., 2010*), is a feasible option. In all of these scenarios, however, there is the need for a robust method that enables estimation of enrichment of the transplanted cells with tumor-specific T cells based on the T cell receptor (TCR) repertoire without prior knowledge of TCR specificities.

The ongoing adaptive immune response is often driven by groups of T cell clones with highly homologous TCR sequences (clusters) that recognize the same epitopes (*Pogorelyy et al., 2019*; *Dash et al., 2017*; *Glanville et al., 2017*; *Pogorelyy et al., 2018*). However, clusters of highly similar TCRs also abundantly arise from the so-called 'public' variants having high probability of being generated in the course of V(D)J recombination (*Venturi et al., 2011*; *Elhanati et al., 2018*). ALICE approach evaluates the number of 'neighbors' relative to the baseline expectation from V(D)J recombination statistics, and retains the node clonotypes of the TCR clusters with higher numbers of neighbors than expected by a null model of recombination (*Pogorelyy et al., 2019*; *Dash et al., 2017*; *Glanville et al., 2017*; *Pogorelyy et al., 2018*). This approach is highly efficient to capture clonotypes involved in the current immune response from a single repertoire snapshot, and does not require longitudinal data collection (*Pogorelyy et al., 2019*; *Dash et al., 2017*; *Glanville et al., 2017*; *Pogorelyy et al., 2018*; *Pogorelyy and Shugay, 2019*).

Here, we have employed ALICE-based cluster analysis to identify groups of TCR clonotypes involved in the anti-tumor immune response. We demonstrate that this approach successfully pinpoints known tumor-associated antigens (TAAs)-specific TCRs among TIL repertoires in HLA-A*02 melanoma patients. Furthermore, we find that the number of cluster-related clonotypes and the proportion of the bulk TIL repertoire that they occupy grow significantly after anti-PD-1 immunotherapy. We next investigate the TCR content in sorted CD4$^+$CD39$^+$PD1$^+$ and CD8$^+$CD39$^+$PD1$^+$ TILs, and show that these subsets are prominently enriched for TCR clusters, a substantial fraction of which consists of tumor-specific TCRs. These results provide a rationale for focusing on CD39$^+$PD1$^+$ TILs in adoptive cancer therapy. Finally, we show that repertoire analysis facilitates optimization of TIL culturing conditions, and allows estimation of the extent of tumor-specific T cell enrichment in cultured donor cells and sorted TAA-activated T cells. Altogether, our findings strongly support the use of cluster TCR analysis as a powerful tool with practical applications in clinical ACT.

## Results

### TIL clusters include TAA-specific TCRs and grow after immunotherapy

We first analyzed published TIL TCR beta chain (TCRβ) repertoires obtained before and after anti-PD-1 immunotherapy for two cohorts comprising 21 and 8 patients with cutaneous melanoma (*Riaz et al., 2017*; *Tumeh et al., 2014*). Using the ALICE algorithm (*Pogorelyy et al., 2019*), we were able to identify clusters of convergent TCRβ clonotypes in all patients. The number of cluster-related clonotypes

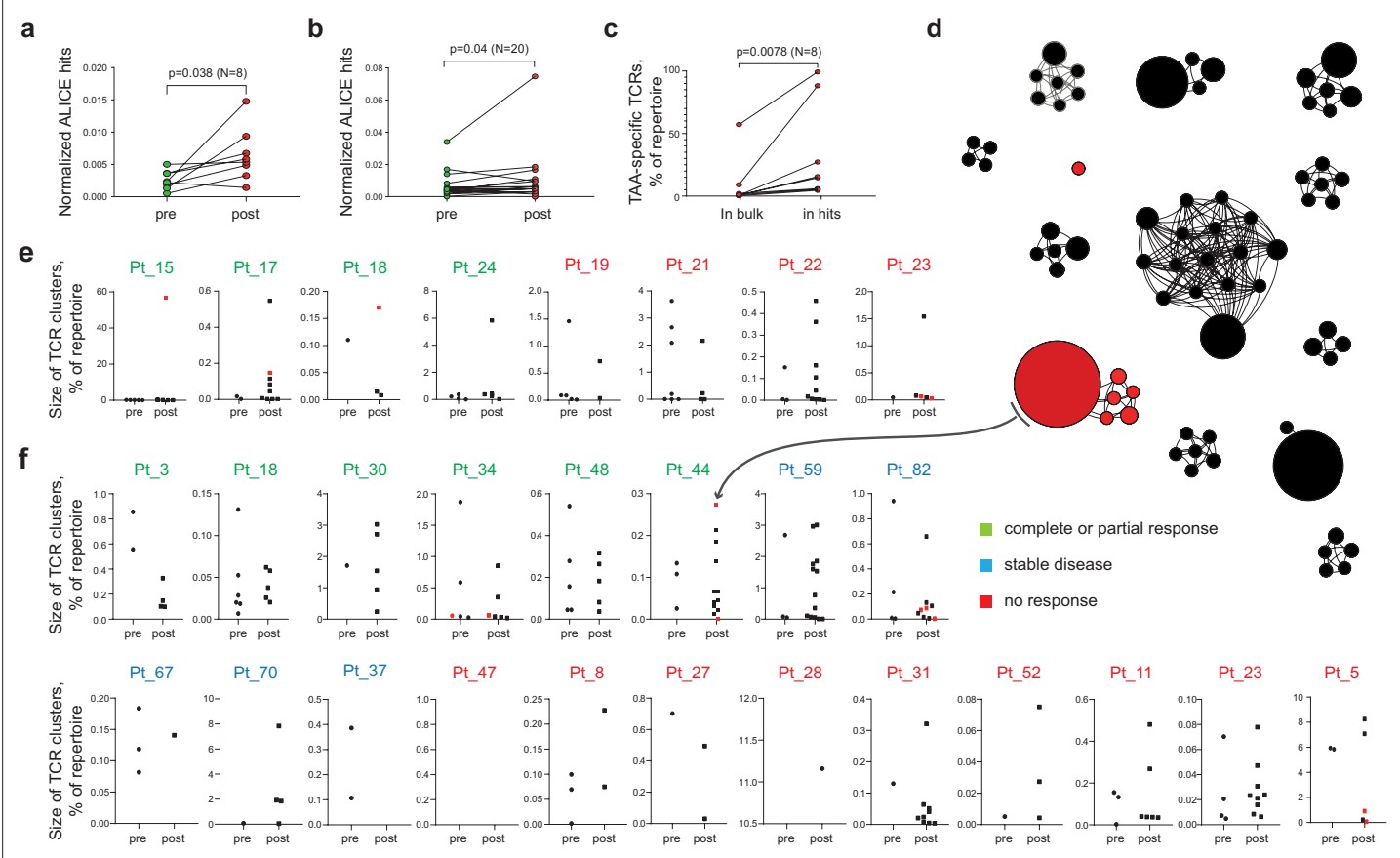

**Figure 1.** Melanoma tumor-infiltrating lymphocyte (TIL) T cell receptor (TCR) clusters before and after immunotherapy. (**a**, **b**) Normalized counts of cluster-related TCRβ clonotypes from metastatic melanoma samples before and after anti-PD-1 therapy from datasets published in (**a**) (*Riaz et al., 2017*) and (**b**) (*Tumeh et al., 2014*). (**c**) Cumulative frequency of VDJdb-matched tumor-associated antigen (TAA)-specific clonotypes within the whole repertoire (in bulk) and within cluster-related clonotypes (ALICE hits) of patients that have at least one VDJdb-matched cluster (N = 8 patients) from the two published datasets. (**d**) TCRβ clusters from patient pt44 (*Riaz et al., 2017*). Each dot represents individual TCRβ clonotype. Dot size is proportional to the clonotype frequency in the bulk TCRβ repertoire. VDJdb-matched TAA-specific clonotypes are colored in red. (**e**, **f**) TCRβ clusters before and after therapy for each patient from (**e**) (*Riaz et al., 2017*) and (**f**) (*Tumeh et al., 2014*). One dot corresponds to one cluster. Y axis shows the size of corresponding clusters (e.g. cumulative frequency of clonotypes within a cluster, % of the whole repertoire). VDJdb-matched TAA-specific clusters are colored in red. N = number of biological replicates in each group. Data in (**a**) were analyzed with paired t-test; b, c were analyzed with the Wilcoxon test.

significantly increased after therapy in both cohorts (p = 0.019 and 0.038, respectively; *Figure 1a and b*), which might reflect treatment-dependent expansion of convergent antigen-specific TILs. At the same time, no statistically significant differences were observed between responders, stable disease, and progressive disease groups.

A VDJdb database (*Shugay et al., 2018*; *Bagaev et al., 2020*) search identified cluster-related clonotypes that are highly similar or identical to known TCRβ variants specific to the HLA-A*02-restricted melanoma-associated antigens Melan-A (Melan-A$_{aa26-35}$ - ELAGIGILTV) and NY-ESO-1 (NY-ESO-1$_{aa157-165}$ - SLLMWITQC) in 20% and 50% of patients from the *Riaz et al., 2017*; *Tumeh et al., 2014*, cohorts, respectively. Cluster-related clonotypes were enriched for TAA-specific TCRβ variants compared to the bulk TCR repertoire, indicating that cluster analysis can capture clonotypes involved in an ongoing anti-tumor immune response (*Figure 1c and d*). *Figure 1d* shows identified TCR clusters for one of the patients after immunotherapy. Summary for the count and size of TCR clusters before and after immunotherapy for each patient is shown in *Figure 1e and f*.

Notably, the HLA genotypes of the patients for these two cohorts were unknown. Since VDJdb currently includes limited diversity of HLA contexts, we believe that a much higher proportion of cluster-related clonotypes will be assigned to TAA specificities with the accumulation of TCR specificity data from more diverse HLA contexts.

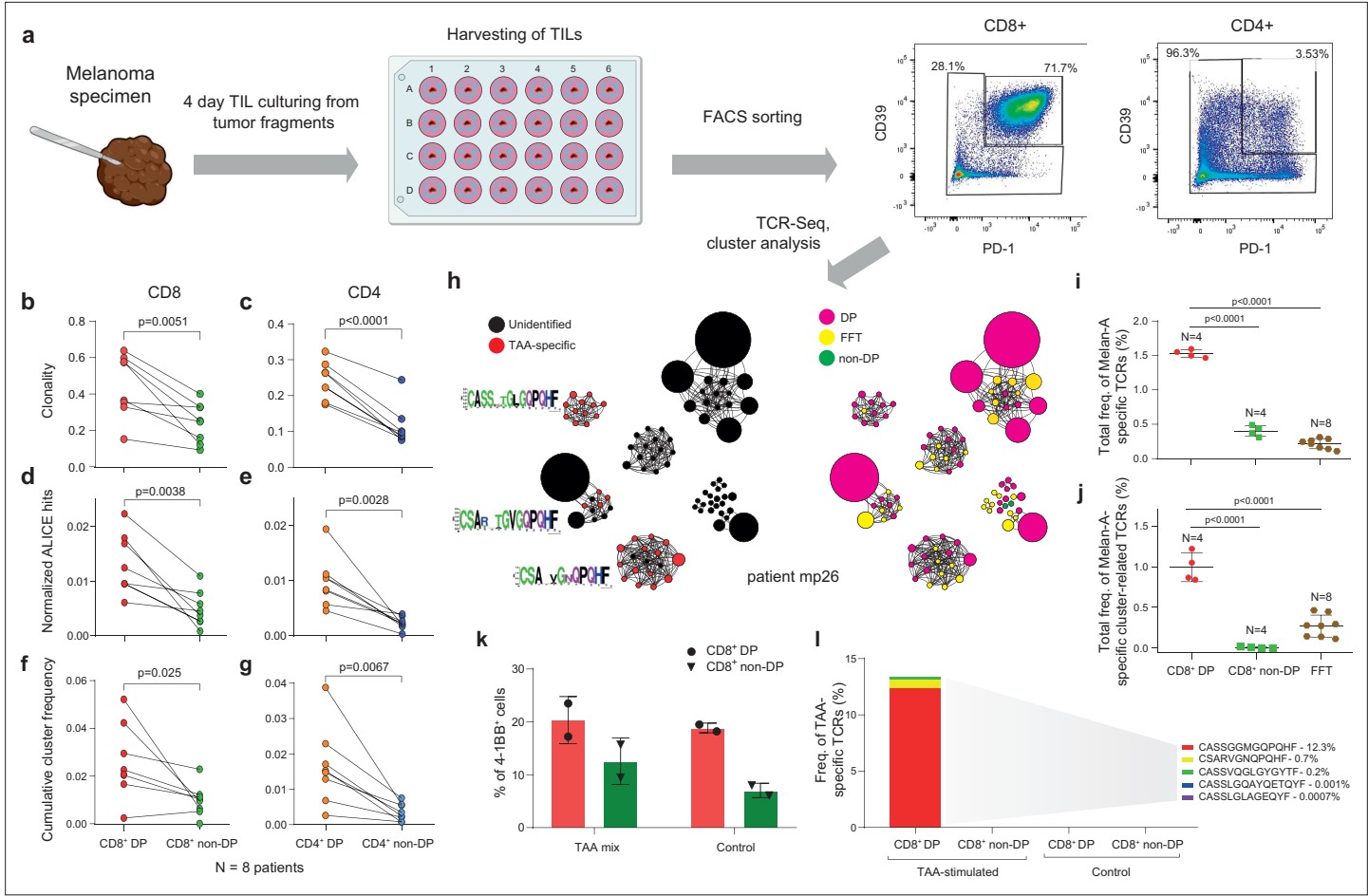

**Figure 2.** T cell receptor (TCR) clusters in CD39⁺PD1⁺ tumor-infiltrating lymphocytes (TILs). (**a**) The experimental workflow. (**b–g**) TCRβ repertoire analysis for CD8⁺ (**b, d, f**) and CD4⁺ (**c, e, g**) double-positive (DP) and non-DP TIL subsets sorted from metastatic lymph nodes of eight melanoma patients. Panels show repertoire clonality calculated as [1 − Normalized Shannon-Wiener index] (**b, c**), normalized counts (**d, e**), and cumulative frequency of cluster-related clonotypes, that is, total weight of all clusters as a proportion of TCRβ repertoire (**f, g**). Paired t-test. (**h**) TCRβ clusters identified in repertoires obtained from fresh-frozen tumor (FFT) samples, and sorted CD8⁺ DP and non-DP TILs for HLA-A*02 patient mp26. VDJdb-matched TAA-specific clusters are colored in red. ( **i, j**) Cumulative frequency of (**i**) VDJdb-matched TAA-specific clonotypes and (**j**) VDJdb-matched TAA-specific cluster-related clonotypes within CD8⁺ DP, CD8⁺ non-DP, and FFT TCRβ repertoires of patient mp26. One-way ANOVA, Bonferroni multiple comparisons correction. (**k, l**) Proportion of CD137⁺ cells among CD8⁺ T cells (**k**) and proportion of VDJdb-matched TAA-specific clonotypes in sorted CD137⁺CD8⁺ T cells (**l**) in DP and non-DP TILs from patient mp26 that were cultured and re-stimulated with TAA-loaded or control dendritic cells.

The online version of this article includes the following figure supplement(s) for figure 2:

**Figure supplement 1.** (**a**) Fluorescence-activated cell sorting (FACS) gating for sorting of CD8⁺ and CD4⁺ CD39⁺PD-1⁺ double-positive (DP) and non-DP T cells from briefly cultured melanoma tumor-infiltrating lymphocytes (TILs) and single-cell suspensions prepared from tumor samples (**b,c**).

**Figure supplement 2.** Characteristics of sorted tumor-infiltrating lymphocytes (TILs).

## CD39⁺PD1⁺ TILs are enriched with clonal, convergent, and tumor-specific TCRs

It was previously reported that CD39⁺ and PD-1⁺ TIL subsets can be enriched for tumor-specific T cells (*Gros et al., 2014*; *Inozume et al., 2010*; *Balança et al., 2021*; *Li, 2020*; *Thommen et al., 2018*; *Simoni et al., 2018*; *Pauken et al., 2021*; *Zhu et al., 2021*). To verify whether there is concurrent enrichment with convergent TCR clusters, we performed fluorescence-activated cell sorting (FACS) of CD39⁺PD-1⁺ (double-positive [DP]) and non-DP CD4⁺ and CD8⁺ T cells from TILs freshly isolated from lymph node metastases from eight melanoma patients (*Figure 2a*, *Figure 2—figure supplement 1a*). Overall, TIL composition was skewed toward greater prevalence of CD4⁺ T cells, although cells with the DP phenotype were more prevalent among CD8⁺ TILs compared to CD4⁺ cells (*Figure 2—figure supplement 2a,b*). TCRβ repertoire analysis revealed increased clonality and cluster enrichment for

both CD4$^+$ and CD8$^+$ DP TILs compared to the corresponding non-DP subsets (*Figure 2b–g*), with greater clonality among CD8$^+$ TILs than CD4$^+$ TILs regardless of immune checkpoint expression status (*Figure 2—figure supplement 2c,d*).

For TILs obtained from HLA-A*02-positive patient mp26, with BRAF$^{wt}$ melanoma, a VDJdb search identified three TCR clusters that included A*02-Melan-A$_{aa26-35}$-specific clonotypes (*Figure 2h*). Clonotypes matching to Melan-A-specific VDJdb entries were prominently enriched within the CD8$^+$ DP subset (about 1.5% of repertoire), compared to fresh-frozen tumor (FFT) tissue and to the CD8$^+$ non-DP subset (*Figure 2i*). Most of these clonotypes belonged to TCR clusters, and such Melan-A-specific cluster-related clonotypes constituted about 1% of CD8$^+$ DP subset repertoire (*Figure 2j*).

Similar results were obtained for another BRAF$^{wt}$ HLA-A*02-positive patient, pt41 (*Figure 2—figure supplement 1b,c*), where TCR repertoires of CD8$^+$ DP and CD8$^+$ non-DP subsets were compared. We concluded that the CD39$^+$PD1$^+$ fraction is enriched for large and convergent T cell clones that are involved in an ongoing tumor-specific immune response, a substantial portion of which are detectable via cluster TCR analysis.

To functionally confirm our findings, we used the CD137 (4-1BB) upregulation assay. Sorted DP and non-DP TIL subsets from patient mp26 were cultured for 2 weeks and stimulated with autologous monocyte-derived dendritic cells loaded with a TAA peptide mix. CD8$^+$CD137$^{high}$ subsets were subsequently quantified with flow cytometry and sorted for TCRβ library preparation.

As shown in *Figure 2k*, the proportion of CD137$^{high}$ cells was higher in cultured DP cells, but we found no difference between T cells stimulated by TAA-loaded or control dendritic cells. At the same time, TCRβ repertoire analysis revealed that the CD137$^{high}$ fraction of TAA-stimulated CD8$^+$ DP—but not non-DP or control DP—cells was enriched with known Melan-A-specific clonotypes (*Figure 2l*). These clonotypes included a TAA-reactive TCRβ variant, CSARVGNQPQHF-TRBV20-TRBJ1-5, which was previously detected in cluster analysis of non-cultured CD8$^+$ DP cells, and variant CASSGGMGQPQHF-TRBV19-TRBJ1-5, which is homologous to another cluster. TAA-specific clonotypes cumulatively occupied ~13% of the CD137$^{high}$ fraction of the cultured and TAA-activated DP TILs. These results underscore the importance of TCR repertoire analysis of responding cells, even in the apparent absence of a quantifiable difference between antigen and control conditions, and demonstrate that CD137 marker analysis on its own is insufficiently informative.

## TCR cluster analysis facilitates optimization of TIL culturing conditions

We next investigated the effect of distinct TIL culture conditions on the expansion of tumor-reactive T cells, including concentration of IL-2 (stimulates expansion of both conventional T cells and T$_{reg}$), and presence of IL-21 (plays a key role in the development and maintenance of memory CD8$^+$ T cells, inhibits T$_{reg}$ proliferation), anti-PD-1 antibody (blocks inhibitory interaction of T cells' PD-1 with PD-L1), and IFNγ (activates antigen presentation and supports type 1 immune response). Four distinct TIL culture conditions included: IL-2$_{high}$, IL-2$_{low}$/IL-21, IL-2$_{low}$/IL-21/anti-PD-1, and IL-2$_{low}$/IL-21/anti-PD-1/IFNγ (*Figure 3—figure supplement 1*), where the anti-PD-1 agent employed was nivolumab and the concentration of IL-2 in the low and high conditions was 100 and 3000 IU/mL, respectively. For each condition, we analyzed TCRβ repertoires of TILs independently cultured from 12 tumor fragments collected from patient mp26. TCR clusters identified from all samples were joined and visualized along with the non-cultured FFT, DP CD8$^+$, and non-DP CD8$^+$, repertoires (*Figure 3a*). As a readout, we used the following:

i. normalized count of cluster-related clonotypes (*Figure 3b*),
ii. cumulative proportion of the repertoire occupied by Melan-A-specific TCRβ clusters (clusters predominantly comprising VDJdb-defined Melan-A-specific clonotypes) (*Figure 3c*), and
iii. the number of differentially expanded clonotypes compared to pan-activating IL-2$_{high}$ culture conditions, and the proportion of such expanded clonotypes that were also initially detected among CD8$^+$ DP TILs (*Figure 3d*).

The IL-2$_{low}$/IL-21/anti-PD-1 combination yielded the greatest number of cluster-related clonotypes and the highest cumulative proportion of Melan-A-specific clusters out of all culture conditions, as well as compared to initial non-cultured FFT samples (*Figure 3b and c*). Addition of IFNγ did not further enhance the expansion of potentially tumor-specific clones.

We utilized edgeR (*Robinson et al., 2010*) software, which was initially designed for differential gene expression analysis, to identify clonotypes that were significantly expanded in tumor fragment

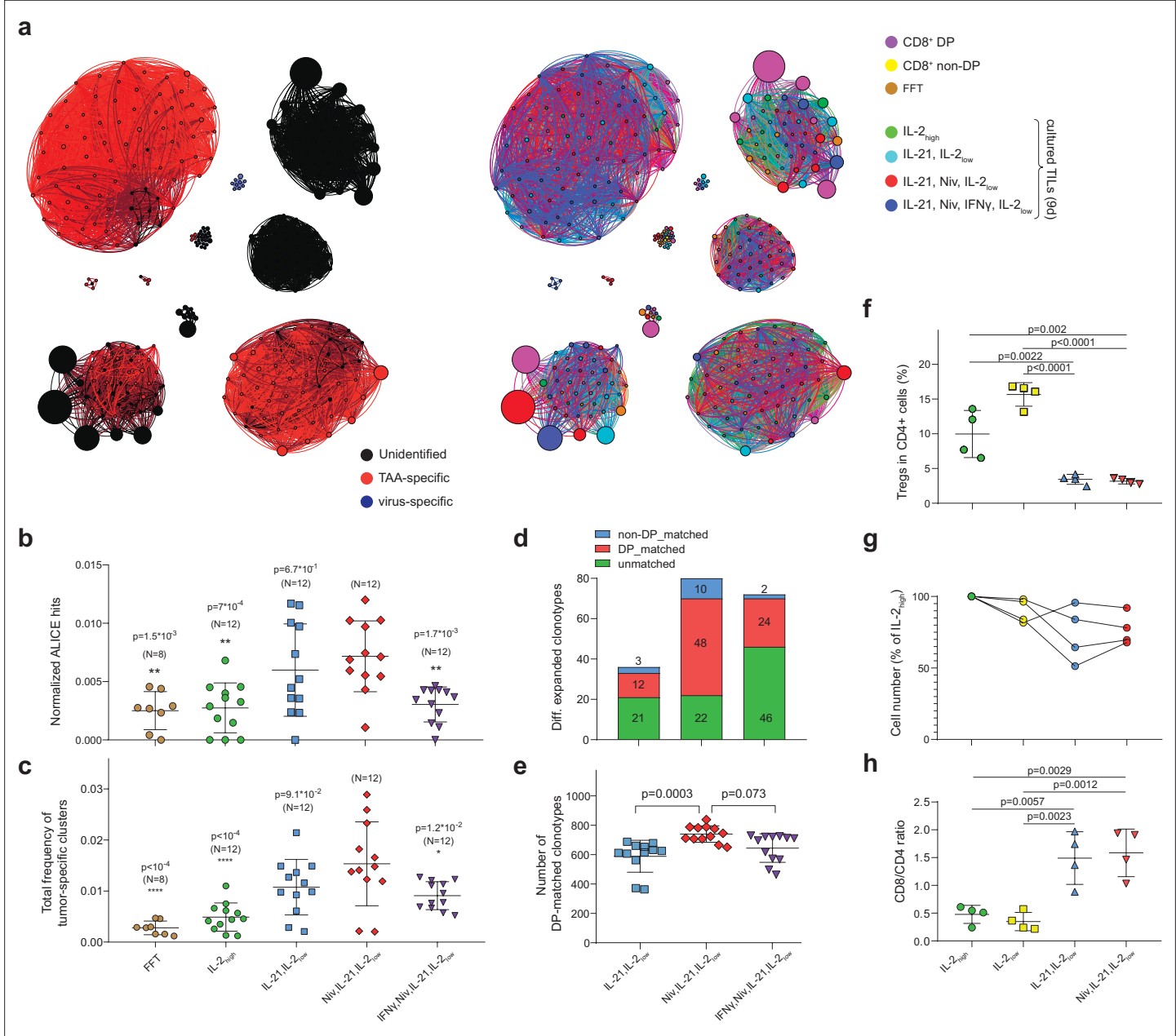

**Figure 3.** Influence of culture conditions on tumor-infiltrating lymphocytes (TILs) and T cell receptor (TCR) clusters. (**a**) Combined TCRβ clusters for fresh-frozen tumor (FFT), CD8+ double-positive (DP), CD8+ non-DP, and TILs cultured in four different conditions from metastatic tumor sample of patient mp26. Left: colors indicate VDJdb-defined clonotype specificities. Right: colors indicate the sample of origin. (**b–e**) TCRβ repertoire analysis for patient mp26 TILs cultured in four distinct conditions. Panels show (**b**) normalized counts of cluster-related TCRβ clonotypes, (**c**) proportion of the repertoire occupied by Melan-A-specific TCRβ clusters, (**d**) count and subset matching of clonotypes preferentially expanded compared to the IL-2high condition, and (**e**) the count of CD8+ DP matched clonotypes among the top 1500 clonotypes. N = number of separate tumor fragments. (**f**) Fraction of regulatory T cells (Tregs) (CD25+CD127-FoxP3+) among CD4+ T cells in cultured TILs from patient mp41 in each culture condition. (**g**) TIL counts relative to IL-2high culture conditions; n = 4 patients. (**h**) CD8+/CD4+ ratio for TILs from patient mp41 in each culture condition. Data in b ,c, f, g were analyzed with one-way ANOVA, Bonferroni multiple comparisons correction, with each condition compared to the IL-21/IL-2low/anti-PD-1 condition. Data in the panel (**e**) was analyzed with Kruskal-Wallis test, Dunn's multiple comparison test, with each group compared to IL-21/IL-2low/anti-PD-1.

The online version of this article includes the following figure supplement(s) for figure 3:

**Figure supplement 1.** Expansion of tumor-infiltrating lymphocyte (TIL) subtypes in different cultivation conditions.

cultures in the presence of IL-21 and IL-2$_{low}$ (either with or without nivolumab and IFNγ) compared to classical pan-activating IL-2$_{high}$ culture conditions. The IL-2$_{low}$/IL-21/anti-PD-1 combination yielded the highest number of reproducibly expanded clonotypes, 60% of which were detected among initial non-cultured CD8$^+$ DP TILs (*Figure 3d*, *Figure 3—figure supplement 1b*). The overall count of such CD8$^+$ DP-matched clonotypes was also highest for this combination (*Figure 3e*). These results show the positive influence of PD-1 inhibition on the proliferative potential of CD8$^+$CD39$^+$PD-1$^+$ T cells. We also noted a slight increase in the number of CD8$^+$ non-DP matched clonotypes which were expanded in these same conditions—10 clonotypes, compared to three clonotypes in IL-2$_{low}$/IL-21 without nivolumab (*Figure 3d*)—which can be explained by nivolumab-dependent expansion of PD-1$^+$CD39$^-$ T cells.

## IL-2$_{low}$/IL-21/anti-PD-1 culture conditions stimulate TIL proliferation without T$_{reg}$ expansion

T$_{reg}$s are known to hamper anti-cancer immune responses elicited by ex vivo expanded TILs (*Melief et al., 2017*). This immunosuppression may be overcome by CD25$^+$ T cell depletion (*Li et al., 2010*). Alternatively, T$_{reg}$ expansion may be suppressed by IL-21 (*Santegoets et al., 2013*). To more comprehensively characterize the distinct TIL culture conditions described above, we compared their capacity to support or suppress T$_{reg}$ expansion and overall T cell expansion. Predictably, IL-2 alone yielded the highest proportion of T$_{reg}$ (CD4$^+$CD25$^{high}$CD127$^{low}$FoxP3$^{high}$ cells) cells. In line with prior findings (*Bonnet et al., 2016*), IL-2$_{low}$ created preferential conditions for selective T$_{reg}$ expansion, while the addition of IL-21 drastically reduced the proportion of T$_{reg}$s among CD4$^+$ TILs (*Figure 3f*, *Figure 3—figure supplement 1c,d*).

IL-21 was previously demonstrated to promote the expansion of T cells with a memory phenotype (*Allard et al., 2007*). We also noted a significant increase in the number of memory phenotype (*Carrette and Surh, 2012*) CD127$^{high}$ cells among CD4$^+$ lymphocytes in IL-21$^+$ conditions regardless of the presence or absence of nivolumab (*Figure 3—figure supplement 1e*). Remarkably, CD8$^+$ TILs displayed significant CD127$^{high}$ enrichment only upon simultaneous introduction of IL-21 and nivolumab (*Figure 3—figure supplement 1f*). These results reveal the synergistic action of IL-21 and disruption of PD-1-dependent signaling by nivolumab on expansion of CD8$^+$ memory T cells.

Regarding overall proliferative potential, the highest T cell count was evident for IL-2$_{high}$ conditions, although we observed comparable numbers in the IL-2$_{low}$ cultures (*Figure 3g*). The presence of IL-21 stifled IL-2-dependent TIL proliferation, as previously reported for human CD4$^+$ T cells (*Ferrari-Lacraz et al., 2008*). On the other hand, IL-21 was favorable for expansion of CD8$^+$ TILs, whereas IL-2 alone favored CD4$^+$ cell growth (*Figure 3h*).

## Discussion

Here, we demonstrate that rational TCR clustering can be used to identify tumor-specific T cell clones and estimate their relative enrichment among TILs. Starting with published TCR repertoire data from melanoma tumors, we identified clusters of convergent TCR clonotypes, which increased in numbers and total frequency after anti-PD-1 immunotherapy. We next found significant enrichment of convergent TCR clusters in PD1$^+$CD39$^+$ subpopulations of both CD4$^+$ and CD8$^+$ TILs, which were previously shown to be enriched in terms of tumor reactivity (*Li, 2020*; *Simoni et al., 2018*; *Kortekaas et al., 2020*). These findings are further supported by the data we obtained for cases where we know the HLA context, as well as some TCR clonotypes of interest and their cognate antigens. A VDJdb database search successfully identified TCR variants specific to TAA antigens in HLA-A*02-positive patients, where approximately half of the clusters could be matched to known Melan-A-specific sequences.

The cumulative frequency of such Melan-A specific clusters within the CD8$^+$ DP population was only slightly lower than the total frequency of all identified Melan-A specific TCRs, indicating that our approach was able to identify most of the high-frequency tumor-specific clonotypes.

It should be noted that Melan A-specific T cells have unusually high frequency in melanoma patients, and even in healthy individuals with HLA-A*02 allele (*Pittet et al., 1999*; *Przybyla et al., 2019*; *Alanio et al., 2010*). Also, Melan A-specific T cell clones in some cases represent predominant population in melanoma TILs (*Ramirez-Montagut et al., 2000*) and are present at high frequency in melanoma-infiltrated lymph nodes (*Romero et al., 1998*). This may mean that using Melan A-specific clonotypes

as a model is overestimating the sensitivity of our approach. On the other hand, in our data, as well as in public datasets we used in this paper, there are other clusters of comparable size that are not identified as Melan A-specific, which means that there are other specificities with similar behavior.

It should be also noted that TCR clusters are an essential feature of a convergent immune response that involves several homologous clonotypes. For those cases where a single T cell clone dominates in response to a particular antigen and/or homologous 'neighbors' are absent due to the very low probability of convergent TCR generation (*Dash et al., 2017*), cluster analysis may miss some of the tumor-reactive TCRs. Nevertheless, our results demonstrate the overall power of this approach, which is applicable in those situations where specific antigens are unknown.

We describe one potential implementation of our approach by using it to optimize conditions for ex vivo TIL expansion. In particular, TILs cultured in IL-21$^+$ conditions demonstrated the highest number of cluster-related TCRs, which is indicative of a more prominent influence of antigen-driven TCR selection. We speculate that IL-21 exerts its influence in our culture system both at the antigen-presentation stage (as MHCI- and, to a lesser extent, MHC II-restricted antigen presentation by tumor cells occurs while tumor fragments are cultured ex vivo) (*Gagnon et al., 2008*) and at the expansion stage, where IL-21 selectively promotes the expansion of non-T$_{reg}$ populations. The consequences of in vitro PD-1 blockade are less well understood. The in vitro application of an anti-PD1 antibody was previously shown to induce IFNγ and TNFα secretion by PD1-expressing CD8$^+$ TILs (*Park et al., 2019*). In our work, we found a greater increase in abundance of cluster-related and tumor-specific TCRs for cells cultured in IL-21$^+$/anti-PD-1 conditions compared to IL-21$^+$ alone.

We expect that this straightforward TCR repertoire-based approach to estimation of TIL enrichment with tumor-reactive clones will accelerate clinical development of adoptive T cell therapy. At the R&D stage, this approach could greatly facilitate the selection of optimal TIL subsets and optimization of TIL culturing conditions and downstream enrichment procedures. And within clinical pipelines, such analyses should make it possible to estimate tumor-reactive TIL abundance at the level of individual tumor samples, both before and after culturing and/or enrichment. As the field of T cell therapy advances toward the ability to determine the clonal specificities and phenotypes that can effectively fight each individual patient's tumor, methods for grouping convergent TCRs that respond to the same tumor antigens will become an essential part of the toolbox for rationally designed T cell therapy.

## Materials and methods

### Key resources table

| Reagent type (species) or resource | Designation | Source or reference | Identifiers | Additional information |
|---|---|---|---|---|
| Other | Human AB serum | PanBiotech, Germany | P30-2901 | Cell culture medium component |
| Peptide, recombinant protein | IL-2 (Ronkoleukine) 1,000,000 IU/mL | BiotechSpb, Russia | – | 100–3000 IU/mL |
| Other | RPMI-1640 | PanEco, Russia | C310 | Cell culture medium |
| Chemical compound, drug | HEPES, pH 7.2, 1 M | PanEco, Russia | Φ134 | 25 µM/mL |
| Chemical compound, drug | Penicillin/streptomycin | Gibco, USA, Waltham, Massachusetts | 10378016 | 1:100 |
| Chemical compound, drug | Gentamicin | Gibco, USA, Waltham, Massachusetts | 15750–037 | 1:5000 |
| Other | Non-essential amino acids mix | Gibco, USA, Waltham, Massachusetts | 11140–035 | 1:100 |
| Other | GlutaMAX | Gibco, USA, Waltham, Massachusetts | 35050–061 | 1:100 |
| Chemical compound, drug | β-Mercaptoethanol | Gibco, USA, Waltham, Massachusetts | 31350010 | 1:1000 |
| Chemical compound, drug | Sodium pyruvate | Gibco, USA, Waltham, Massachusetts | 11360–070 | 1:100 |
| Other | RLT cell lysis buffer | Qiagen, The Netherlands | 160046763 | Sample:RLT ratio >1 : 3.5 |
| Peptide, recombinant protein | IL-21 | SCI-STORE, Russia | PSG260-10 | 50 ng/mL |

*Continued on next page*

*Continued*

| Reagent type (species) or resource | Designation | Source or reference | Identifiers | Additional information |
|---|---|---|---|---|
| Antibody | Nivolumab (Opdivo 10 mg/mL) | Bristol-Myers Squibb, USA, New York City, New York | NDC 00003-3772-11 | 1:500 |
| Commercial assay or kit | Anti-CD3/CD28 Dynabeads | Thermo Fisher Scientific, USA | 11131D | |
| Other | Ficoll-Paque Plus | GE Healthcare, USA, Chicago, Illinois | GE17-1440-03 | For PBMC isolation |
| Other | SepMate-50 tubes | StemCell Technologies, Canada | 86450 | For PBMC isolation |
| Commercial assay or kit | Anti-CD14 MicroBeads | Miltenyi Biotec, Germany | 130-050-201 | |
| Other | X-Vivo-15 medium | Lonza, Switzerland | BE02-060F | Cell culture medium for dendritic cells |
| Peptide, recombinant protein | IL-4 | PeproTech, Germany | 200–04 | 400 IU/mL |
| Peptide, recombinant protein | GM-CSF | PeproTech, Germany | 300–03 | 800 IU/mL |
| Peptide, recombinant protein | PepTivator Melan-A/MART-1 | Miltenyi Biotec, Germany | 130-094-597 | 1:100 |
| Peptide, recombinant protein | PepTivator gp100/Pmel | Miltenyi Biotec, Germany | 130-094-449 | 1:100 |
| Peptide, recombinant protein | PepTivator MAGE-A3 | Miltenyi Biotec, Germany | 130-095-384 | 1:100 |
| Chemical compound, drug | Prostaglandin $E_2$ | Sigma-Aldrich, USA, St. Louis, Missouri | P0409 | 1 µg/mL |
| Peptide, recombinant protein | IL-1β | PeproTech, Germany | 200-01B | 10 ng/mL |
| Peptide, recombinant protein | TNF-α | PeproTech, Germany | 300–01A | 25 ng/mL |
| Other | AIM-V serum-free medium | Gibco, USA, Waltham, Massachusetts | 12055–091 | Serum-free cell culture medium |
| Antibody | Mouse monoclonal CD4-BV510 (RPA-T4) | BioLegend, Germany | cat 300546 | FACS, flow cytometry, 5 µL/test |
| Antibody | Mouse monoclonal CD8-Alexa-647 (SK1) | BioLegend, Germany | cat 344726 | FACS, flow cytometry, 5 µL/test |
| Antibody | Mouse monoclonal CD127-APC-Cy7 (IL-7Rα, A019D5) | BioLegend, Germany | 351348 | FACS, flow cytometry, 5 µL/test |
| Antibody | Mouse monoclonal CD25-PE (IL-2Rα) | Beckman Coulter, USA, Brea, California | A07774 | FACS, flow cytometry, 10 µL/test |
| Antibody | Mouse monoclonal CD39-FITC (eBioA1) | eBioscience, USA, Santa Clara, California | 11-0399-42 | FACS, flow cytometry, 5 µL/test |
| Antibody | Mouse monoclonal CD137-PE (4B4-1) | BioLegend, Germany | 309804 | FACS, flow cytometry, 5 µL/test |
| Antibody | Mouse monoclonal CD25-APC (BC96) | BioLegend, Germany | 302610 | FACS, flow cytometry, 5 µL/test |
| Antibody | Mouse monoclonal PD-1-BV421 (CD279, EH12.2H7) | BioLegend, Germany | 329920 | FACS, flow cytometry, 5 µL/test |
| Antibody | Mouse monoclonal CD127-Alexa488 (A019D5) | BioLegend, Germany | 351314 | FACS, flow cytometry, 5 µL/test |
| Antibody | Mouse monoclonal CD8-BV421 (SK1) | BioLegend, Germany | 344748 | FACS, flow cytometry, 5 µL/test |
| Antibody | Mouse monoclonal anti-FoxP3-PE (236 A/E7) | BD Biosciences, USA, Franklin Lakes, New Jersey | 560852 | Flow cytometry, 5 µL/test |
| Commercial assay or kit | TRIzol reagent | Invitrogen, USA, Carlsbad, California | 15596026 | |
| Commercial assay or kit | RNeasy mini kit | Qiagen, The Netherlands | 74104 | |
| Commercial assay or kit | Qubit RNA HS Assay Kit | Thermo Fisher Scientific, USA, Waltham, Massachusetts | Q32855 | |
| Commercial assay or kit | Human RNA TCR Multiplex kit | MiLaboratories Inc, USA, Sunnyvale, California | – | |

## Patients

All clinical samples were acquired from the NN Blokhin National Medical Research Center of Oncology in accordance with protocol MoleMed-0921, approved by the ethical committee on January 30, 2020.

All patients involved in the study were diagnosed with metastatic melanoma and signed an informed consent prior to collection of their biomaterial. Most experiments were performed on freshly resected metastatic lymph nodes obtained during surgery. Genotyping for BRAF$^{V600E}$ was performed at the NN Blokhin National Medical Research Center of Oncology, Pathomorphology Department. From each patient enrolled in this study, we also obtained 20–30 mL of peripheral blood before the surgery. Patient information is provided in *Supplementary file 1*.

## Brief TIL culture

Freshly resected tumor specimens were dissected into fragments measuring 1–3 mm in each dimension. Several fragments were frozen in liquid nitrogen for further cDNA library preparation and HLA typing. Individual fragments were seeded into the wells of a 24-well tissue culture plate with 2 mL of complete T cell cultivation medium (CM) supplemented with 5% heat-inactivated human AB serum (PanBiotech, Germany) and 1000 IU/mL IL-2 (Ronkoleukine, BiotechSpb, Russia). CM consisted of RPMI-1640 (PanEco, Russia), 25 µM/mL HEPES, pH 7.2 (PanEco, Russia), 100 IU/mL penicillin, 100 µg/mL streptomycin, 10 µg/mL gentamicin, 1× non-essential amino acids mix, 1× GlutaMAX, β-mercaptoethanol (0.55 µM) (all from Gibco, Thermo Fisher Scientific, USA, Waltham, Massachusetts), and 110 µg/m sodium pyruvate. On the fourth day of cultivation, TILs were harvested, filtered through a 70 µm mesh, stained with fluorophore-labeled antibodies, and FACS sorted. For the generation of TCR repertoire libraries, T cells were sorted directly into the RLT cell lysis buffer (QIAGEN, The Netherlands) and stored at –80°C until used for RNA isolation. For functional assays, TILs were sorted into 1.5 mL Eppendorf tubes with 0.5 mL RPMI-1640. Live-sorted T cells were seeded into wells of 96-well cell culture plate at $10^6$ cells/mL in CM and cultured for at least 5 days. Here, CM was supplemented with 1000 IU/mL IL-2, 50 ng/mL IL-21 (SCI-STORE, Russia), 20 µg/mL nivolumab (Bristol-Myers Squibb, USA, New York City, New York), and 10% autologous patient-derived serum. Half of the media was replaced three times a week. One day before the functional assays, all the media was replaced with fresh CM without interleukins or nivolumab.

## Expansion of bulk TILs

Fragments of tumor specimens from each of four melanoma patients (mp26, -32, -34, and -42) were plated into separate wells of a 24-well tissue culture plate (1 fragment per well, 1 plate per patient) with basic RPMI-1640-based CM and 5% heat-inactivated human AB serum. Bulk TILs were expanded from tumor fragments in the following different conditions (6 wells/condition) for 9 days: (1) IL-2$_{high}$ (3000 IU/mL), (2) IL-2$_{low}$ (100 IU/mL), (3) IL-21 (25 ng/mL) + IL-2$_{low}$, (4) nivolumab (20 µg/mL) + IL-21 + IL-2$^{low}$, and (5) IFNγ (100 µg/mL) + nivolumab + IL-21 +IL-2$_{low}$. TILs from patient mp26 were cultivated in conditions 1, 3, 4, and 5; TILs from patients mp32, -34, and -42 were cultured in conditions 1–4. Half of the medium was replaced on days 3 and 5 with the same medium as was used for initial plating. On day 9 of cultivation, TILs from each well were counted and lysed in RLT buffer at $5 \times 10^5$ cells/mL density for preparation of TCR repertoire libraries.

## Expansion of sorted T cells

FACS-sorted T cells were expanded using non-specific TCR-dependent stimulation with anti-CD3/CD28 Dynabeads (Thermo Fisher Scientific, USA, Waltham, Massachusetts). Beads were added into the CM on the next day after seeding the cells, with 2 µL of bead solution per $10^5$ cells. Sorted TILs were expanded for 2 weeks. Beads were magnetically removed upon achieving desired cell numbers. Before co-cultivation experiments, cells were allowed to 'rest' for 1 day in interleukin-free CM supplemented with 10% autologous patient-derived serum (i.e., 'resting' medium).

## PBMC isolation

Peripheral blood mononuclear cells (PBMCs) were derived from patients' blood samples using gradient centrifugation with Ficoll-Paque Plus (GE Healthcare). Briefly, 18 mL of whole blood were diluted to 50 mL volume with sterile 1× PBS. Diluted blood was layered over the Ficoll-Paque solution in 50 mL SepMate tubes (StemCell Technologies, Canada), with 25 mL of diluted blood per tube. SepMate tubes were centrifuged for 20 min at 1200× *g* with brake off. Afterward, buffy coats were collected and washed two times with 50 mL of sterile PBS.

## Monocyte-derived dendritic cells cultivation

Autologous dendritic cells were generated as described in *Bacher et al., 2016*. Briefly, CD14$^+$ cells (monocytes) were isolated from patients' PBMCs with a magnetic enrichment procedure using anti-CD14 MicroBeads (Miltenyi Biotec, Germany). Then, monocytes were seeded into the wells of 24-well tissue culture plates at $5 \times 10^5$ cells/well. X-Vivo-15 medium (Lonza, Switzerland) with 400 U/mL IL-4 and 800 U/mL GM-CSF was used for the differentiation of monocytes. On the fourth day of cultivation, the medium was renewed, and dendritic cells were loaded with the mix of melanoma TAA peptides (PepTivator Melan-A/MART-1, gp100/Pmel, and MAGE-A3 human; Miltenyi Biotec) at a concentration of 600 nM each. The next day, loaded DCs were matured using 1 µg/mL PGE, 10 ng/mL IL-1β, and 25 ng/mL TNF-α. Following 24 hr of maturation, DCs were harvested and used for co-cultivation with T cells.

## CD137 antigen-specific activation assay

FACS-sorted PD1$^+$CD39$^+$ (DP) and non-DP populations after expansion and 2 days 'rest' without IL-2 were co-cultured with antigen-loaded autologous DCs at a ratio of 10:1 T cells:DCs. The co-CM consisted of 1:1 CM plus AIM-V serum-free medium (Gibco, USA, Waltham, Massachusetts) supplemented with 50 ng/mL IL-21. Both CD137$^{high}$ and CD137$^{low}$ T cells were lysed with RLT buffer for further RNA isolation and TCR library construction. The frequency of CD137$^{high}$ cells was measured for both CD4$^+$ and CD8$^+$ TILs. Antigen-specific activation was measured as a ratio of CD137$^{high}$ T cell frequencies in TILs co-cultured with antigen-loaded vs. unloaded DCs.

## Flow cytometry

For FACS of PD1$^+$CD39$^+$ TIL subpopulations, cells were stained with the following antibodies: CD4-BV510 (RPA-T4), CD8-Alexa-647 (SK1), CD127-APC-Cy7 (IL-7Rα, A019D5), PD-1-BV421 (CD279, EH12.2H7) (BioLegend, Germany), CD25-PE (IL-2Rα, Beckman Coulter, USA, Brea, California), CD39-FITC (eBioA1) (eBioscience, USA, Santa Clara, California). Briefly, cells were pelleted, resuspended in the staining solution with fluorescent antibodies, and incubated for 1 hr at 4°C in the dark. Next, the cells were washed and resuspended in PBS at an approximate density of $5 \times 10^6$ cells/mL. T cells of interest were sorted on a FACS Aria III (BD Biosciences, USA, Franklin Lakes, New Jersey). Data analysis was performed with FlowJo software (FlowJo, LLC, FlowJo, USA, Ashland, Oregon). To avoid antigen-specific T$_{reg}$-dependent T cell suppression in further functional assays, we identified T$_{reg}$s as CD4$^+$CD25$^+$CD127$^-$ cells (*Yu et al., 2012*) and sorted them separately. For functional assays, we sorted T cells into CD39$^+$ PD-1$^+$ DP and non-DP (CD39 or PD-1 single-positive and double-negative) subpopulations, with CD4$^+$ and CD8$^+$ cells together. For TCR library construction, CD4$^+$ DP and CD8$^+$ DP cells, as well as corresponding non-DP populations, were sorted separately in order to evaluate individual properties of their TCR repertoires. For the CD137 activation assay, T cells were stained with fluorescently labeled CD4-BV510 (RPA-T4), CD8-Alexa-647 (SK1), CD137-PE (4B4-1) (BioLegend, Germany) antibodies.

## Nuclear staining for FoxP3

First, T cells were stained for surface markers with CD4-BV510 (RPA-T4), CD25-APC (BC96), CD127-Alexa488 (A019D5) (BioLegend, Germany) for 30 min at 4°C in the dark. TILs from patient mp41 were also stained with CD8-BV421 (SK1, BioLegend, Germany) to estimate the CD8/CD4 ratio in various cultivation conditions. Next, the cells were washed and resuspended and processed with the eBioscience Intracellular Fixation & Permeabilization Buffer Set according to manufacturer's instructions with minor alterations. Briefly, stained cells were fixed with Fixation Reagent for 45 min at 4°C in the dark. Then, the fixed cells were washed three times in 1 mL of Permeabilization Reagent, resuspended in 50 µL of Permeabilization Reagent and blocked for 15 min in 2% human serum. Next, cells were stained with 2 µL of anti-FoxP3-PE (236 A/E7) (BD Biosciences, USA, Franklin Lakes, New Jersey) for 1 hr in Permeabilization buffer at 4°C in the dark, washed, and analyzed with a Navios flow cytometer (Beckman Coulter, USA, Brea, California). Finally, the fraction of T$_{reg}$s (CD4$^+$ CD25$^{high}$ CD127$^{low}$ FoxP3$^{high}$) out of total CD4$^+$ T cells was determined.

## HLA typing

For patients mp39, mp41, mp42, and mp44, we have only checked for the presence of HLA-A*02. Aliquots of PBMCs or TILs from these patients were stained with anti-HLA-A*02-PE (BD7.2) (BD

Biosciences, USA, Franklin Lakes, New Jersey) antibody and analyzed with a Navios flow cytometer. Other patients (*Supplementary file 1*) were HLA typed using NGS at the Center for Precision Genome Editing and Genetic Technologies for Biomedicine (Moscow). For mp26, the following HLA alleles were identified: A*02:01, A*26:01, B*27:05, B*73:01, C*02:02, C*15:05.

## RNA isolation and TCR library preparation

RNA from fresh-frozen tumor fragments was isolated using TRIzol reagent (Invitrogen). RNA from RLT-lysed cells was extracted using the RNeasy mini kit (Qiagen) according to the manufacturer's protocol. RNA concentration was measured with the Qubit RNA HS Assay Kit (Thermo Fisher Scientific, USA, Waltham, Massachusetts). No more than 500 ng of total RNA was used for cDNA synthesis. cDNA libraries were generated using the Human RNA TCR Multiplex kit (MiLaboratories), according to the manufacturer's protocol. We aimed to achieve coverage of 20 paired-end reads per cell for sorted and cultivated TIL populations, and approximately $2 \times 10^6$ reads per tumor sample fragment. Sequencing was performed using the Illumina NextSeq platform ($2 \times 150$ bp read length).

## Analysis of TCR repertoires sequencing data

TCRseq data was analyzed with MiXCR software (MiLaboratories) in order to extract TCRβ CDR3 clonotypes. VDJtools software was used for processing of MiXCR output TCR repertoire data, calculation of TCR repertoire diversity, and pre-processing of TCR repertoires (i.e., pooling of joined TCR repertoires and down-sampling of TCR repertoires) (*Shugay et al., 2015*).

## Cluster analysis of TIL TCRβ repertoires

This analysis was performed using the ALICE algorithm (*Pogorelyy et al., 2019*). We selected clonotypes with read count >1 (removed singletons) to exclude undercorrected erroneous TCR variants that could potentially create false neighbors of abundant clonotypes, distorting the ALICE hit identification process. Briefly, TCRβ repertoire data files were converted into VDJtools tabular format. For each repertoire, we selected columns with amino acid CDR3 sequence, TRBV, TRBJ, clonotype frequency, and read count. These pre-processed TCR repertoire tables were used as an input for the ALICE algorithm. $P_{gen}$ of amino acid sequences was estimated using Monte Carlo simulation. For each VJ pair, 5 million TCR sequences were simulated, and 20 iterations of the algorithm were performed. Based on $P_{gen}$, ALICE infers the number of highly similar CDR3 sequences (up to 1 amino acid mismatch) for each clonotype. Under the expectation that antigen-driven clonal selection doesn't occur, TCRs with high probability of generation are expected to have a high number of neighbors, while for low-probability TCRs there should be low number or no neighbors. Based on this assumption, clonotypes with significantly higher numbers of neighbors than expected from the recombination model (Benjamini-Hochberg-adjusted p < 0.001) are expected to have come through convergent antigen-specific selection and expansion. The algorithm output is formed as a list of these significant results referred to as 'ALICE hits'.

The number of ALICE hits strongly depends on the initial variability of the TCR repertoire (*Figure 2—figure supplement 1d*). To account for this, 'Normalized ALICE hits' metric was calculated by dividing the number of ALICE hits by the number of clonotypes in initial input. ALICE code can be found at https://github.com/pogorely/ALICE (*Pogorelyy and Green, 2019*).

To visualize the resultant clusters of convergently selected TCRs, we used the igraph function (*Csardi, 2005*). This function utilizes the de Bruijn graph method to calculate the distance between amino acid sequences of CDR3 regions. It creates a graph file in GML format where each node represents an individual TCR clonotype and the distance between nodes is proportional to the difference between CDR3 amino acid sequences. Edges connected nodes representing TCR clonotypes with Hamming distance ≤1. Graphs were visualized using Gephi 0.9.2 network analysis platform (*Bosler et al., 2009*). The size of the node represents the frequency of the corresponding TCR clonotype. Upon construction of composite graphs including clonotypes from multiple TCR repertoires, identical clonotypes from different TCR repertoires were displayed by separate nodes.

## Matching cluster-related TCR clonotypes to VDJdb

We annotated TCR repertoire data using the VDJdb database of TCR with known specificity (*Shugay et al., 2018*). We assumed that TCRs of interest had the same specificity as TCRs from the database

if: (1) CDR3 regions of compared TCRs differed no more than by one central amino acid substitution, (2) substituted amino acids belonged to the same group based on their R properties (polar, aliphatic, aromatic, positively/negatively charged), and (3) HLA restriction of TCR clonotypes from the database matched one of the patient's HLA alleles, if known.

TCR clusters consisting predominantly of TAA-specific clonotypes, but not clonotypes of other specificities, were considered TAA-specific as a whole. VDJdb-unmatched members of the TAA-specific clusters were deemed to possess the same specificity as the whole cluster based on structural similarity to VDJdb-matched clonotypes and were included in the subsequent analysis.

## Analysis of differentially expanded clonotypes with edgeR software

We used a statistical approach implemented in the edgeR (*Robinson et al., 2010*) package to identify TCRβ clonotypes that were significantly expanded in bulk TILs of patients mp26 and mp34. We implemented edgeR for comparison of TCR repertoires of TILs cultivated in experimental conditions 2–4 (described above) and TILs expanded in IL-2$_{high}$ conditions. Six and four biological replicate samples of each cultivation setting were used for the analysis of TCR repertoires from mp26 and mp34, respectively. TCR clonotypes were deemed expanded if the false discovery rate adjusted p value was < 0.01 and the log$_2$ fold-change was >1.

## Statistical analysis

Statistical analysis was performed using GraphPad Prism 8.0 (GraphPad Software Inc, USA, San Diego, California). All data was reported as mean ± SD. The Shapiro-Wilk test was used for normality estimation in all cases. Names of statistical tests and numbers of biological replicas in each comparison group are provided in the figure legends.

## Acknowledgements

We thank the Center for Precision Genome Editing and Genetic Technologies for Biomedicine (Moscow) for the genetic research methods, and personally V Cheranev and D Korostin for HLA typing of patient tumor material. We thank Valeria V Kriukova for the help with data analysis. We thank Skoltech Genomics Core Facility for performing high-throughput sequencing of mp41 libraries (Skoltech Life Sciences Program grant for the use of shared facilities, received by M Goncharov). We are grateful to Michael Eisenstein for his valuable help in editing the manuscript. Funding: Supported by the grant of the Ministry of Science and Higher Education of the Russian Federation № 075-15-2020-807.

## Additional information

### Funding

| Funder | Grant reference number | Author |
| --- | --- | --- |
| Ministry of Science and Higher Education of the Russian Federation | 075-15-2020-807 | Dmitriy M Chudakov |

The funders had no role in study design, data collection and interpretation, or the decision to submit the work for publication.

### Author contributions

Mikhail M Goncharov, Data curation, Formal analysis, Methodology, Writing – original draft; Ekaterina A Bryushkova, Investigation, Methodology, Writing – original draft; Nikita I Sharaev, Vadim Karnaukhov, Data curation, Methodology; Valeria D Skatova, Data curation, Formal analysis, Methodology; Anastasiya M Baryshnikova, George V Sharonov, Investigation, Methodology; Maria T Vakhitova, Methodology; Igor V Samoylenko, Investigation, Resources, Writing – original draft; Lev V Demidov, Conceptualization, Project administration, Writing – original draft; Sergey Lukyanov, Conceptualization, Funding acquisition; Dmitriy M Chudakov, Conceptualization, Funding acquisition, Investigation, Methodology, Resources, Supervision, Visualization, Writing – original draft, Writing

- review and editing; Ekaterina O Serebrovskaya, Conceptualization, Data curation, Methodology, Supervision, Visualization, Writing – original draft, Writing - review and editing

### Author ORCIDs
Dmitriy M Chudakov  http://orcid.org/0000-0003-0430-790X

### Ethics
Human subjects: All clinical samples were acquired from the N.N. Blokhin National Medical Research Center of Oncology in accordance with protocol MoleMed-0921, approved by the ethical committee on 30 Jan 2020. All patients involved in the study were diagnosed with metastatic melanoma and signed an informed consent prior to collection of their biomaterial.

### Decision letter and Author response
Decision letter https://doi.org/10.7554/eLife.77274.sa1
Author response https://doi.org/10.7554/eLife.77274.sa2

---

## Additional files

### Supplementary files
- Supplementary file 1. Patients and clinical characteristics.
- MDAR checklist

### Data availability
TCR repertoires have been deposited on: https://figshare.com/projects/ Pinpointing_the_tumor-specific_T-cells_via_TCR_clusters/125284.

The following dataset was generated:

| Author(s) | Year | Dataset title | Dataset URL | Database and Identifier |
|---|---|---|---|---|
| Goncharov M | 2021 | Pinpointing the tumor-specific T-cells via TCR clusters | https://figshare.com/ projects/Pinpointing_ the_tumor-specific_ T-cells_via_TCR_ clusters/125284 | figshare, 125284 |

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
