## [Editor Report]

This manuscript presents a computational approach to identify T-cells that can mount an immune response against tumors. The authors examine the presence of clusters of T cells with similar sequence as a surrogate for tumor antigen-specific responses. The identification of tumor-specific responses within the background of bystander T cell infiltration is an area of great current interest. This study provides solid support that T cell sequence clustering can be used to identify tumor-specific responses in vivo and in vitro.

---

## [Decision Letter]

**Decision letter after peer review:**

Thank you for submitting your article "Pinpointing the tumor-specific T-cells via TCR clusters" for consideration by *eLife*. Your article has been reviewed by 3 peer reviewers, and the evaluation has been overseen by a Reviewing Editor and Satyajit Rath as the Senior Editor. The following individuals involved in review of your submission have agreed to reveal their identity: Yuval Elhanati (Reviewer #1); Benny Chain (Reviewer #2); Giulio Isacchini (Reviewer #3).

Essential revisions:

1. Please include a description of the ALICE algorithm so the manuscript becomes self-sufficient.

2. ALICE is a powerful method since it can find reactive T cells without a background before sample. But for the scenarios in the paper, a before sample usually exists, which can help identify the reactive clusters and clones.

Are these samples used to inform ALICE? If not, it should be discussed how they can used to strengthen the analysis.

3. Please better describe the differences between different T-cell subgroups, which are sorted based on receptor such as IL2, IL21, etc. These descriptions should make the paper more accessible to the broader audience.

4. In the first result section, and in figure 1, it is shown that normalized ALICE hits increase following treatment. But it's not clear how this effect differs between responders and non-responders. This should be added as a figure or at least described in the text.

5. Melan A is a rather unusual TAA, because of the high precursor frequency of melan A specific T cells. Please include a discussion on this point in the Discussion section.

6. Please clarify the figure captions and descriptions (in the text) by thoroughly addressing all comments from reviewer #3.

7. Please clarify the online methods by thoroughly addressing the comments from reviewer #4.

*Reviewer #1 (Recommendations for the authors):*

– In the last paragraph of the introduction, a comma or space seems to be missing between CD8^+^ and CD39+PD1+, at least if indeed the CD39+PD1+ refers also to the CD4^+^ cells.

– Also in the same paragraph, the acronym TAA is used, but it's not defined outside of the abstract.

– ALICE is a powerful method since it can find reactive T cells without a background before sample. But for the scenarios in the paper, a before sample usually exist, that can help identify the reactive clusters and clones. Is it used in some way in ALICE? If not, it should at least be discussed how it can used to strengthen the analysis.

– Various cytokines and receptors such as IL2, IL21, CD137 and CD39 are used to separate T cells of interest, often without describing their function and role. For the wider audience in *eLife*, these should be described in context, so the different groups of T cells used are clear.

– In the first Results section, and in figure 1, normalized ALICE hits are shown to increase following treatment. But it's not clear how this effect changes between responders and non-responders. This should be added as a figure or at least described in the text.

– The last section in the results seems to have little relation to the main analysis method in the paper, and maybe should be omitted.

– In the methods section "Cluster analysis of TIL TCRβ repertoires", the following sentence is unclear – "we normalized the number of hits between samples based on the number of the top-frequency input clonotypes."

*Reviewer #2 (Recommendations for the authors):*

This study is on the whole convincing and interesting. I felt that the figures, while aesthetically pleasing, could have been explained in more detail, and were somewhat hard to follow.

1. I would like to know more about the set of matching TCRs in VDJdb – how many, against what antigens, are the annotated TCRs themselves highly clustered.

2. For example, in 1C, in the clusters, does this mean that all TCRs were also found in VDJdb? Or does this mean that there was only a tiny cluster and so one hit was close to 100%? What is the individual red dot in panel 1 d? What exactly does the y axis in f mean? Is this the proportion of TILs which fall in an ALICE cluster?

3. A bit more detail on running the ALICE pipeline would be helpful. I couldn't see anything about this in Methods.

4. A bit more explanation of the y axes in general would be helpful. What exactly is the calculation for normalised ALICE hits, etc.

5. I like the point in Figure 2 that phenotype is sometimes not enough, but needs sequencing alongside. This is an important message to the field.

6. I don't quite understand the gigantic clusters of TCRs in Figure 3, and why they all seem to be red. This comes back to point 1 I think. I don't quite understand how stimulating a population in vivo gives rise to these huge clusters, all within 2 aa of each other? Maybe I am not understanding this figure – again a bit more detail saying what exactly is being done and shown would hugely improve the impact.

7. Melan A is a rather unusual TAA, because of the high precursor frequency of melan A specific T cells. Might be worth discussing this in a bit more detail in the Discussion.

In summary, this is a very interesting and important piece of work. But more thought on clarifying what is being done, and how it is being shown would hugely improve the study and increase the number of readers.

*Reviewer #3 (Recommendations for the authors):*

Some more explanations (in the main text as well) on the Alice algorithm would be useful for those that are not familiar with the approach.

Moreover, also in the Online Methods some clarifications are needed:

– It is mentioned that the number of Alice hits depends on the initial variability of the repertoire: can you show it with an SI figure?

– Where does the normalization with respect to top-frequency clones comes from? Is an heuristic approach or has it a proper basis? Why does it work?

– Is the default pgen model used? Why not inferring a pgen model for each of the patients?

– The Alice algorithm should work with clusters of hamming distance one. Why visualizing clusters of hamming distance up to 2?

---

## [Author Response]

Essential revisions:1. Please include a description of the ALICE algorithm so the manuscript becomes self-sufficient.

We added a brief description to the Introduction, and also expanded in the Methods section.

2. ALICE is a powerful method since it can find reactive T cells without a background before sample. But for the scenarios in the paper, a before sample usually exists, which can help identify the reactive clusters and clones.Are these samples used to inform ALICE? If not, it should be discussed how they can used to strengthen the analysis.

We discuss this point in detail in our answer to the corresponding Reviewer 2 question.

3. Please better describe the differences between different T-cell subgroups, which are sorted based on receptor such as IL2, IL21, etc. These descriptions should make the paper more accessible to the broader audience.

We added brief explanations about each marker or cytokine.

4. In the first result section, and in figure 1, it is shown that normalized ALICE hits increase following treatment. But it's not clear how this effect differs between responders and non-responders. This should be added as a figure or at least described in the text.

No difference between responders and non-responders. Added to the text, discussed in our answer to the corresponding Reviewer 2 question.

5. Melan A is a rather unusual TAA, because of the high precursor frequency of melan A specific T cells. Please include a discussion on this point in the Discussion section.

Please see our answer to the corresponding Reviewer 3 question.

6. Please clarify the figure captions and descriptions (in the text) by thoroughly addressing all comments from reviewer #3.

Worked on this, thank you.

7. Please clarify the online methods by thoroughly addressing the comments from reviewer #4.

Worked on this as well, thank you.

Reviewer #1 (Recommendations for the authors):– In the last paragraph of the introduction, a comma or space seems to be missing between CD8^+^ and CD39+PD1+, at least if indeed the CD39+PD1+ refers also to the CD4^+^ cells.

Corrected, thank you.

– Also in the same paragraph, the acronym TAA is used, but it's not defined outside of the abstract.

Corrected, thank you.

– ALICE is a powerful method since it can find reactive T cells without a background before sample. But for the scenarios in the paper, a before sample usually exist, that can help identify the reactive clusters and clones. Is it used in some way in ALICE? If not, it should at least be discussed how it can used to strengthen the analysis.

Indeed, the beauty of the ALICE approach is that – since it allows to identify convergent TCRs involved in the current response – it allows to get the useful information “from single repertoire snapshots”. In the past, we successfully used ALICE to identify autoimmunity-related TCR clusters, as well as Yellow-fever specific clusters, all from single snapshots (https://journals.plos.org/plosbiology/article?id=10.1371/journal.pbio.3000314).

Here, we use non-DP and bulk (FFT) repertoires as background for comparison. On Figure 3b,c, the FFT represent non-cultured TCR repertoires, so essentially these are “before” samples if you compare with the cultured cells on the same panels. Figure 3a also includes non-cultured DP CD8^+^ and non-DP CD8^+^ repertoires that also represent “before” samples.

Furthermore, on Figure 3d,e, where we analyse clonotypes that preferentially expanded in presence of IL-21 and IL-2_low_ (with or without nivolumab and IFNɣ) compared to pan-activating IL-2_high_ conditions, we then check the presence of such clonotypes in initial non-cultured DP CD8^+^ and non-DP CD8^+^ repertoires – that is, again, in “before” samples. Thus, we actually use the background before samples, not in the ALICE approach but to evaluate the informativeness of the obtained clusters.

We worked on the text to make it clear, thank you for the note.

– Various cytokines and receptors such as IL2, IL21, CD137 and CD39 are used to separate T cells of interest, often without describing their function and role. For the wider audience in eLife, these should be described in context, so the different groups of T cells used are clear.

We added brief description of cytokines used for TILs culturing:

“We next investigated the effect of distinct TIL culture conditions on the expansion of tumor-reactive T cells, including concentration of IL-2 (stimulates expansion of both conventional and regulatory T cells), and presence of IL-21 (plays a key role in the development and maintenance of memory CD8^+^ T cells, inhibits T_reg_ proliferation), anti-PD-1 antibody (blocks inhibitory interaction of T cells’ PD-1 with PD-L1), and IFNɣ (activates antigen presentation and supports type 1 immune response). Four distinct TIL culture conditions included…”

We also briefly added:

“Certain cell-surface markers of ongoing and chronic activation such as PD-1, CD39, CD69, CD103, or CD137 may also help to delineate T cell subpopulations (typically CD8^+^) that are enriched for clonally-expanded tumor-reactive T cells…”

– In the first Results section, and in figure 1, normalized ALICE hits are shown to increase following treatment. But it's not clear how this effect changes between responders and non-responders. This should be added as a figure or at least described in the text.

Thank you for the note. We checked but found no differences. The following was added to the text:

“The number of cluster-related clonotypes significantly increased after therapy in both cohorts (p = 0.019 and 0.038, respectively; Figure 1a,b), which might reflect treatment-dependent expansion of convergent antigen-specific TILs. At the same time, no statistically significant differences were observed between responders, stable disease, and progressive disease groups.”

– The last section in the results seems to have little relation to the main analysis method in the paper, and maybe should be omitted.

For a part of the readership, this manuscript is not only about bioinformatics, but also about practical aspects of TIL-based therapy.

Furthermore, this section of results shows that the T cell expansion condition that showed highest enrichment with tumor-specific TCR clusters as shown by ALICE, at the same time yields preferable T cell subsets composition – in terms of low Treg, high memory, and high CD8^+^ T cells proportions.

Thus, we believe that this part is a valuable piece of the manuscript that should preferably remain in the main text.

– In the methods section "Cluster analysis of TIL TCRβ repertoires", the following sentence is unclear – "we normalized the number of hits between samples based on the number of the top-frequency input clonotypes."

Thank you for the note, it was not neither clear nor formally correct. Now added Supplementary Figure 1d and changed to a more clear description:

“The number of ALICE hits strongly depends on the initial variability of the TCR repertoire (Supplementary Figure 1d). To account for this, “Normalized ALICE hits” metric was calculated by dividing the number of ALICE hits by the number of clonotypes in initial input.”

Reviewer #2 (Recommendations for the authors):This study is on the whole convincing and interesting. I felt that the figures, while aesthetically pleasing, could have been explained in more detail, and were somewhat hard to follow.1. I would like to know more about the set of matching TCRs in VDJdb – how many, against what antigens, are the annotated TCRs themselves highly clustered.

Different ALICE clusters had different counts of VDJdb-matching clonotypes, that matched variable numbers of VDJdb entries. For example, ALICE clusters from Figure 2h in the manuscript matched to a set of MLANA-specific VDJdb entries:

**Author response image 1. sa2fig1:** 

Here you can see the graphs and logos for TAA-specific clusters from Figure 3a:

**Author response image 3. sa2fig3:** 

Notably, all matches had specificity to the same antigen and same HLA-restriction, which coincided with the HLA-status of the patient.We have also registered the presence of ALICE cluster, all members of which matched to VDJdb TCRs recognizing NLVPMVATV epitope of pp65 CMV protein in HLA-A*02 context.

**Author response image 4. sa2fig4:** 

There were also some minor clusters that were composed of clonotypes matching VDJdb TCRs with different specificities. We implied that the whole cluster is antigen specific only if all its VDJdb-matched members corresponded to TCRs with the same antigen specificity.

2. For example, in 1C, in the clusters, does this mean that all TCRs were also found in VDJdb? Or does this mean that there was only a tiny cluster and so one hit was close to 100%? What is the individual red dot in panel 1 d? What exactly does the y axis in f mean? Is this the proportion of TILs which fall in an ALICE cluster?

On panel 1с we compare cumulative frequency of TCRs matched to melanoma-specific entries in VDJdb in the total TCR repertoire versus cumulative frequency of such clonotypes among ALICE hits. The purpose of such comparison is to show that ALICE hits are enriched with potentially tumor-reactive clonotypes.

Each dot on panel 1d represents an individual TCR clonotype. Dot size is proportional to the clonotype frequency in the corresponding TCR repertoire. VDJdb-matched TAA-specific clonotypes are colored in red.

Regarding 1e,f panels – each dot represents one TCR cluster. The Y axis shows the size of corresponding clusters (e.g. cumulative frequency of clonotypes within a cluster, % of the whole repertoire).

We worked on the figure legend to make these things more clear.

3. A bit more detail on running the ALICE pipeline would be helpful. I couldn't see anything about this in Methods.

We expanded the Methods section to give more details on ALICE. Some basic description was also added to the Introduction.

4. A bit more explanation of the y axes in general would be helpful. What exactly is the calculation for normalised ALICE hits, etc.

Thank you for the comment, we worked on the figures and text to make these (complicated) things more clear.

5. I like the point in Figure 2 that phenotype is sometimes not enough, but needs sequencing alongside. This is an important message to the field.

Yes, that’s really a critically important message, thank you for the note.

6. I don't quite understand the gigantic clusters of TCRs in Figure 3, and why they all seem to be red. This comes back to point 1 I think. I don't quite understand how stimulating a population in vivo gives rise to these huge clusters, all within 2 aa of each other? Maybe I am not understanding this figure – again a bit more detail saying what exactly is being done and shown would hugely improve the impact.

Clusters in Figure 3 represent convergent clonotypes found in TILs cultured in various cytokine environments, one circle represents 1 clonotype in one condition. Red colour represents clonotypes that were matched to known TAA-specific clonotypes (in VDJdb) allowing one-aa mismatch (Hamming distance ≤1, worked further on this figure), thus red clusters are the ones dominated by clonotypes with identified specificity. We don’t merge clonotypes that come from different samples (corresponding to culture conditions or sorted populations). So the more different variables we analyse, the bigger the clusters will be. Despite this bias, this is a clear way to visualise how culture conditions affect the structure of convergent TCR clusters. For instance, we can see that the top-left cluster is relatively better expanded in IL21+Nivolumab+ culturing conditions.

7. Melan A is a rather unusual TAA, because of the high precursor frequency of melan A specific T cells. Might be worth discussing this in a bit more detail in the Discussion.

We agree that this is a point that needs to be addressed. Added to discussion:

“The cumulative frequency of such Melan-A specific clusters within the CD8^+^ DP population was only slightly lower than the total frequency of all identified Melan-A-specific TCRs, indicating that our approach was able to identify most of the high-frequency tumor-specific clonotypes. It should be noted that Melan A-specific T cells have unusually high frequency in melanoma patients, and even in healthy individuals with HLA-A*02 allele. Also, Melan A-specific T cell clones in some cases represent predominant population in melanoma TILs and are present at high frequency in melanoma-infiltrated lymph nodes. This may mean that using Melan A-specific clonotypes as a model is overestimating the sensitivity of our approach. On the other hand, in our data, as well as in public datasets we used in this paper, there are other clusters of comparable size that are not identified as Melan A-specific, which means that there are other specificities with similar behaviour. "

In summary, this is a very interesting and important piece of work. But more thought on clarifying what is being done, and how it is being shown would hugely improve the study and increase the number of readers.

Thank you! We hope we now improved the manuscript following your very useful comments.

Reviewer #3 (Recommendations for the authors):Some more explanations (in the main text as well) on the Alice algorithm would be useful for those that are not familiar with the approach.

We added general description of ALICE to the Introduction section, and more details to Methods.

Moreover, also in the Online Methods some clarifications are needed:– It is mentioned that the number of Alice hits depends on the initial variability of the repertoire: can you show it with an SI figure?

Yes, thank you for the point. We added the Supplementary Figure 1d that shows this correlation.

– Where does the normalization with respect to top-frequency clones comes from? Is an heuristic approach or has it a proper basis? Why does it work?

Thank you for the note, it was not neither clear nor formally correct. Changed to a more clear description:

“The number of ALICE hits strongly depends on the initial variability of the TCR repertoire (Supplementary Figure 1d). To account for this, “Normalized ALICE hits” metric was calculated by dividing the number of ALICE hits by the number of clonotypes in initial input.”

– Is the default pgen model used? Why not inferring a pgen model for each of the patients?

Indeed, the default VDJ recombination model from Murugan et al. 2012 was used. According to this work, Pgen predicted by proposed default model are overall consistent across population. Moreover, implementation of this model was proven to be valid in seminal ALICE paper.

– The Alice algorithm should work with clusters of hamming distance one. Why visualizing clusters of hamming distance up to 2?

Right, thank you for your note. Should be Hamming distance < = 1, we corrected and re-draw the clusters on Figure 3. On Figures 1 and 2 it was Hamming distance < = 1 initially.